# Evaluation of the Risk of Birth Defects Related to the Use of Assisted Reproductive Technology: An Updated Systematic Review

**DOI:** 10.3390/ijerph19084914

**Published:** 2022-04-18

**Authors:** Dawid Serafin, Beniamin Oskar Grabarek, Dariusz Boroń, Andrzej Madej, Wojciech Cnota, Bartosz Czuba

**Affiliations:** 1Serafin Clinic, 44-100 Gliwice, Poland; 2Department of Gynecology and Obstetrics with Gynecologic Oncology, Ludwik Rydygier Memorial Specialized Hospital, 31-826 Kraków, Poland; bgrabarek7@gmail.com (B.O.G.); dariusz@boron.pl (D.B.); 3Department of Histology, Cytophysiology, and Embryology, Faculty of Medicine, University of Technology, Academy of Silesia, 41-800 Zabrze, Poland; 4Department of Gynaecology and Obstetrics, Faculty of Medicine, University of Technology, Academy of Silesia, 41-800 Zabrze, Poland; 5Department of Pharmacology, Faculty of Medicine, University of Technology, Academy of Silesia, 41-800 Zabrze, Poland; andrzejmadej@o2.pl; 6Department of Women’s Health, Faculty of Health Sciences in Katowice, Medical University of Silesia, 40-752 Katowice, Poland; wojciech.cnota@sum.edu.pl (W.C.); bczuba64@gmail.com (B.C.)

**Keywords:** prenatal diagnostic, assisted reproductive techniques, birth defects, fetal, PRISMA

## Abstract

Fertility problems constitute a serious medical, social, and demographic problem. With this review, we aim to critically appraise and evaluate the existing literature surrounding the risk of birth defects in offspring conceived using techniques based on assisted reproductive technology (ART). Based on searches of the literature in PubMed and ScienceDirect, we obtained a total of 2,003,275 works related to the topic. Ultimately, 11 original papers published in the last 10 years qualified for inclusion in the study. Based on five studies included in this analysis, it was shown that ART significantly increases the risk of congenital malformations in associated newborns. Due to the specifics of given studies, as well as potential confounding risk factors, this influence cannot be ignored. Therefore, considering the information contained in the articles included in this systematic review, it was determined that the risk of birth defects is not directly related to the use of ART itself but also depends on the age of partners, causes of infertility, comorbidities, and the number of fetuses during a pregnancy, as well as many other factors not covered in the literature. It is thus necessary to impress upon infertile couples who wish to have offspring that the use of ART is not risk-free but that the benefits outweigh the risks. Further education in this field, as well as social understanding, is also required.

## 1. Introduction

### 1.1. Fertility and Infertility Problem

Fertility problems constitute a serious medical, social, and demographic problem [1,2]. The World Health Organization defines infertility as a failure to achieve pregnancy after 12 months or more of regular unprotected sexual intercourse (2–4 times a week) [3,4]. Due to ever-growing infertility rates, which presently concern 10–16% of couples worldwide, including about one million in Poland [5], according to the Polish Society of Reproductive Medicine and Embryology (PTMRiE) and the Fertility and Sterility Special Interest Group of the Polish Society of Gynecologists and Obstetrics (SPiN PTGiP), it is recommended that patients who are under 35 years old begin diagnosis and infertility treatment after a year of unsuccessful attempts at becoming pregnant. For women aged 35–40 years, diagnosis and therapy should be considered 6 months after unsuccessful attempts at pregnancy, whereas after 40, as early as directly after a declaration of procreation is made [6,7]. Infertility is defined as a failure to achieve a pregnancy after 1 year of regular monthly sexual intercourse with the aim of reproduction (2–4 times a week) and without the use of any contraceptive measures. Involuntary childlessness is a problem that concerns a high percentage of the population. It is estimated that this phenomenon affects as high as 20% of couples globally; in Poland, the problem of infertility affects approximately one million couples [3,4]. Recently, there has been an increase in the number of couples seeking aid in enlarging their family, which is most likely connected to better access to specialist treatment, modern diagnosis, society becoming richer, and an increase in social awareness [8]. An increasingly common problem with becoming pregnant can also be connected with observed sociocultural changes, which have resulted in the postponement of decisions about maternity until later in life [1], at which point it can be difficult for therapy to be effective and to obtain an adequate reason for this with regard to the treatment used [2]. It is often women who initially decide to seek help in connection with childlessness, agreeing to diagnosis and the commencement of therapy. On the other hand, among men, the dominant approach to the problem of infertility is skepticism because they believe that the only cause of the discussed problem is their partner, and they are negatively predisposed to discuss their own fertility with a specialist. Such a situation is described as the “theft of the woman’s reproductive time” [5] because with increasing age, there are unbeneficial changes in the activity of ovaries and quality of egg cells, in addition to the occurrence of dysfunctions in fissuration processes and an increase in the percentage of spontaneous miscarriages. In men, with increasing age, there are also changes in the hormone concentration profile, including a decrease in the concentration of testosterone with a simultaneous increase in the concentration of testosterone binding protein; however, this connection has not been fully explained to date [9,10,11,12].

### 1.2. Characteristics of Assisted Reproduction Techniques

The term assisted reproduction technology (ART) was introduced in the 1970s. It covers all methods that are used to assist in a successful pregnancy, where ART techniques are used to replace the biological functions connected with procreation. The Polish Society of Reproductive Medicine and Embryology (PTMRiE) and the Polish Society of Gynecologists and Obstetrics (PTGiP) recommend further optimization of ART techniques [6,7,13].

Intrauterine inseminations (artificial insemination) are classified as in vivo methods of introducing semen either of a husband/partner (AIH, artificial insemination by husband) [6] or a donor (AID, artificial insemination by donor) [14] or its components into a woman’s birth canal after laboratory preparation [15].

Insemination is conducted in a natural cycle or after ovulation stimulation, which increases the success rate of the procedure. However, it must be stated that, per recommendations of the PTGP, stimulation is only recommended if no more than three pre-ovulation ovarian follicles are present. The highest effectiveness is observed within three attempts of intrauterine insemination [16,17].

Among the methods available for treating infertility, regardless of its cause, in vitro fertilization is characterized by having the greatest efficiency. Prior to the procedure, a man must conduct a karyotype test and be subject to analysis for microdeletions of the AZF region on the Y chromosome. In the case of women, it is recommended that the ovarian reserve be evaluated to allow for selection of the best possible method for ovary hyperstimulation [18,19].

The process of in vitro fertilization (IVF) can be conducted under in vitro conditions using the classic method, which is based on the introduction of previously prepared sperm cells into egg cells or on a technique of sperm cell microinjection into an egg cell that is selected randomly (ICSI, intracytoplasmic sperm injection) or morphologically (IMSI, intracytoplasmic morphologically selected sperm injection). This method is recommended in infertility connected with the male factor [20].

The choice of the appropriate strategy of ovarian hyperstimulation allows for the growth of many ovarian follicles and, subsequently, for the collection of many mature egg cells. Oocytes obtained as a result of puncture conducted under the control of ultrasound are subject to a further process of fertilization under the conditions of an embryologic laboratory. Then, an intrauterine transfer of one to two embryos is performed after an incubation period of 2–6 days [6].

In order to cause the hyperstimulation of ovaries, drugs that belong to the group of gonadotropin-releasing hormone analogs (short or long protocol) or gonadotropin-releasing hormones (GnRH) are used [21,22].

A critical point in determining the effectiveness of in vitro fertilization is embryo transfer, meaning the transfer of the embryo into the uterus and minimization of the risk of incidence of luteal phase defects [18].

The principal recommendations for IUI and in vitro fertilization are summarized in Table 1.

### 1.3. Prenatal Diagnostic

Prenatal diagnosis includes all tests that may be performed prior to the birth of the child and is a significant achievement in the field of contemporary perinatology. The dynamic development of prenatal diagnosis has been observed within the last 20 years, which is connected with the popularization of techniques of fetal ultrasound imaging (USG), as well as progress in the fields of biochemistry, immunology, immunogenetics, cytogenetics, and molecular biology. Intrauterine fetal imaging creates the possibility of early detection of most developmental defects, which translates to an increase in the success rate of intrauterine treatment or surgical intervention directly after birth. Apart from that, an increase in the survival rate or complete recovery of fetuses with a congenital defect is possible thanks to specialist, comprehensive care facilities of the highest referral level [27,28].

Methods of prenatal diagnosis can be categorized as non-invasive or invasive. The first group includes fetal ultrasound; biochemical testing of the blood sampled from pregnant women for the presence of markers, such as free beta-subunit human chorionic gonadotropin, pregnancy-associated plasma protein A, and a-fetoprotein (AFP); and cell-free DNA testing. On the other hand, invasive testing includes trophoblast biopsy, amniopunction, cordocentesis, and fetoscopy [29].

The principal goal of prenatal diagnosis is to detect all kinds of pathologies in the development of the fetus, as well as irregularities of a genetic nature. It has been confirmed that as the mother’s age increases, the risk of a child being burdened with a genetic and/or congenital development defect also increases. However, it must be understood that the mother’s age is not the only factor that increases the risk of the incidence and development of fetal defects [7,30].

Given the above, the Fetal Medicine Foundation has developed guidelines for prenatal diagnosis in the first trimester of pregnancy based not only on the mother’s age but also on markers of chromosomal aberrations in fetuses [7,31]. One of the newest indicators used to evaluate the risk of incidence of development of defects in fetuses is the marking of cell-free DNA in the blood of pregnant women. This method is characterized by a detection rate of more than 99% with regard to the most commonly occurring trisomy, also showing a low percentage of false-positive results.

The source of free-cell DNA is the placenta, and its detection is possible as early as the fourth week of pregnancy [32]. The fetal genetic material that can be detected in the mother’s blood constitutes 3–6% of the entire extracellular DNA in the mother’s bloodstream. The biological material for analysis is full blood collected from the pregnant woman, from which DNA is then extracted. Due to the fact that until now, no method of complete separation of a mother’s DNA from fetal DNA has been developed, the evaluation is based on the confirmation or exclusion of the nucleotide sequence of genes that are not present in the mother’s blood, e.g., RHD, SRY, and DYS14, with the use of qPCR [32,33].

Therefore, with this review, we aim to critically appraise and evaluate the existing literature surrounding the risk of birth defects after use of assisted reproductive techniques.

## 2. Methods

A systematic search was carried out independently by two investigators using PubMed and ScienceDirect databases (D.S. and B.O.G.) according to the preferred reporting items for systematic reviews and meta-analyses (PRISMA) statement [34]. The following search terms were included: in vitro fertilization, intrauterine inseminations, assisted reproduction techniques, cardiac defect, cardiac defects, non-cardiac defect, non-cardiac defects. The keywords were adapted to the syntax rules of each database. In this systematic review, we included original, peer-reviewed research papers on the risk of developing birth defects in babies conceived with assisted reproductive technology. Case reports were excluded from the review, as were all types of publications other than original papers (reviews, systematic reviews, and meta-analyses) published more than 10 years ago and articles focused on animal studies. In addition, scientific articles published in a language other than English were not included in this systematic review. Filters were used to eliminate work that did not meet our inclusion criteria. In the last stage of our search, the selected articles were assessed in terms of content, and only those works that were actually related to the topic we described ultimately qualified for inclusion.

## 3. Results

In total, based on PubMed and ScienceDirect searches, we obtained 2,003,275 works related to the topic, of which 179,866 were identified in PubMed and 1,823,409 works in ScienceDirect. After the removal of duplicate records (*n* = 14,356) and works that concern animals (*n* = 26,265), 1,988,918 records remained. Given the large number of works found, the criteria of only papers published in the last 10 years and for which full texts were available was instrumental in excluding 1,199,150 works. We also excluded papers written in languages other than English and articles that were not based on original research, as well as systematic reviews and meta-analyses. Ultimately, 11 papers were included in this systematic review. A flowchart of the selection process is presented in Figure 1.

In Table 2, we present the characteristics of the included studies.

Most of the works that qualified for this systematic review are original articles (73%), and 27% were classified by the journal as a cohort study. Considering the inclusion and exclusion criteria we used, most of the works were published in 2017, although the research was conducted in the period of several years preceding the publication year for the work. The included studies were conducted on a relatively large population, considering the cultural context of the ART methods. Therefore, these studies appear to be reliable.

Davies et al. [35], based on a retrospective study based on a comparison of infertility treatment, report using the ART registry of births and terminations with a gestation period of at least 20 weeks or a birth weight of at least 400 g and registries of birth defects. The risk of congenital defects diagnosed before the age of 5 was assessed in a group of women who underwent ART due to infertility, among women who spontaneously became pregnant, among women in the infertility register who became pregnant without the use of ART, as well as in a group of infertile women who were not included in the infertility register. Of the 308,974 births, 2% or 6163 were the result of ART. These authors reported a higher risk of birth defects in children born following ART compared to spontaneous pregnancies (513 defects (8.3%) vs. 17,546 defects (5.8%); odds ratio (OR): 1.47; 95% confidence interval (CI): 1.33 to 1.62), and after adjusting for multiple variables, OR was 1.28 (95% CI, 1.16 to 1.41) with application of ART techniques; for in vitro fertilization (IVF) (165 congenital defects, 7.2%), they were 1.26 (95% CI, 1.07 to 1.48) and 1.07 (95% CI, 0.90 to 1.26), and the odds ratios for ICSI (139 defects, 9.9%) were 1.77 (95% CI, 1.47 to 2.12) and 1.57 (95% CI, 1.30 to 1.90) [35].

Over a period of 6 years, Luke et al. [36] assessed 459,623 Massachusetts women assigned to one of the following three groups: IVF (10,149 women), subfertile (8054 women), or fertile (441,420 women) at risk of complications during pregnancy or birth or having birth defects in the newborn. Based on the conducted study, it was found that the greatest risks for IVF women are uterine bleeding (adjusted risk ratio (ARR) 3.80, 95% CI 3.31, 4.36) and placental complications (ARR 2.81, 95% CI 2.57, 3.08), and for infants, they are very premature delivery (ARR 2.13, 95% CI 1.80; 2.52) and very low birth weight (ARR 2.15; 95% CI 1.80, 2.56). Nevertheless, these authors indicated that women with IVF and those with reduced fertility are older and have more comorbidities compared to women with normal fertility [36].

In a subsequent posting that qualified for this systematic review, Boulet et al. [37], on the basis of 10 years of observation (2000–2010), reported that 64,861 (1.4%) out of 4,618,076 liveborn newborns were conceived using ART. Based on statistical analysis, among newborns conceived with the use of ART, the risk of developing a non-chromosomal defect, after adjusting for maternal characteristics and age, was ARR 1.28; 95% CI, 1.15–1.42 (58.59 per 10,000 newborns for ART, *n* = 389 cases vs. 47.50 per 10,000 newborns for non-ART, *n* = 22,036 cases). The incidence of tracheoesophageal fistula/esophageal atresia (ARR, 1.90; 95% CI, 1.23–2.94) and rectal and colon atresia/stricture (ARR, 1.88; 95% CI, 1.26–282) was higher in ART deliveries compared to non-ART deliveries. In women under 35 years of age, the incidence of Down syndrome was higher in ART versus non-ART deliveries (ARR 1.63; 95% CI 1.05–2.54), but this relationship was not statistically significant (*p* > 0.05). On the other hand, among mothers over 35 years of age, the frequency of chromosomal defects was lower in ART deliveries than in deliveries without ART (ARR 0.66; 95% CI 0.49–0.88). A valuable supplement to the study was the determination of the risk of birth defects among single and multiple pregnancies in the compared groups. In single pregnancies, the risk of birth defects in newborns for ART compared to non-ART neonates was estimated at 63.69 per 10,000, *n* = 218 vs. 47.17 per 10,000, *n* = 21,251 (ARR 1.38; 95% Cl, 1.21–1.59). Among multiple births, the incidence of rectal and colon atresia/stenosis was higher in ART deliveries compared to non-ART deliveries (ARR 2.39; 95% CI 1.38–4.12). A factor significantly increasing the risk of congenital malformations in newborns born after ART is maternal ovulation disorder (ARR, 1.53; 95% CI, 1.13–2.06) [37].

Iwashima et al. [38] decided to assess whether the use of ART affects the incidence of congenital heart disease in the Japanese population. A total of 2716 pregnant women were enrolled in the study, including 399 (14.7%) women pregnant as a result of ART (142 patients received ovulation induction agents (OIA), 56 received AIH, 159 received IVF, and 42 received ICSI). In that prospective study, congenital heart disease was confirmed or ruled out on the basis of two-dimensional echocardiography. In the group of women whose pregnancy resulted from the use of ART, a severe heart defect requiring surgical treatment or leading to death within a year was reported in 5 fetuses, whereas in the group of women whose pregnancy resulted from spontaneous conception, a severe heart defect was reported in 19 fetuses (*p* = 0.892) [38]. A similar study was conducted by Wen et al. [43], who showed a higher incidence of congenital heart defects in pregnancies obtained after ART (223 cases, 2.2%) than in pregnancies obtained without the use of ART (6057 cases, 1.2%; OR 1.82, 95% Cl 1.59–2.09). However, when the risk assessment analysis took into account the potential confounding risk factors, such as mother’s age, income, education, place of residence, smoking, alcohol consumption, drug use, use of folic acid during pregnancy, mental problems, and obesity, the ARR was determined to be 1.70; 95% CI, 1.48–1.95. In addition, it was shown that the risk of congenital heart defects for ART neonates was significantly associated with twin pregnancy and was higher than in a singleton pregnancy (ARR 1.68; 95% CI 1.44–1.92 vs. 1.09; 95% CI, 0.93–1.25) [43].

An interesting study on the risk of congenital malformations in pregnancies resulting from ART is that of Tatsumi et al. [39], who analyzed whether the use of letrozole to stimulate ovulation may affect the occurrence of complications during pregnancy and the risk of developing congenital malformations. Based on the obtained results, it was found that the risk of miscarriage was significantly reduced in the group of women treated with letrozole compared to women who did not use ovulation stimulation (ARR 0.37, 95% CI, 0.30–0.47, *p* < 0.001). In addition, there was no difference in the overall risk of major birth defects between the two groups (letrozole 13 cases; 1.9% vs. natural cycle 34 cases; 1.5%, ARR 1.24, 95% CI, 0.64–2.40, *p* = 0.52). One fetus had anencephaly (pre-term termination of pregnancy), five newborns had ventricular septal defects, two had atrial septal defects, two Down syndrome, one had Edwards syndrome, one was diagnosed with cleft lip without cleft palate, one had congenital hydronephrosis, one had duodenal atresia, one had endocardial cushion defect, and one had hypospadias. The presence of more than one heart defect was observed in 3 out of 12 live births. Importantly, there was no increased risk of congenital anomalies in patients undergoing in vitro fertilization or ICSI [39].

A similar study was conducted by Sene et al. [42], who compared the risk of birth defects for fetuses in the group of pregnant women who received letrozole (105 cases) vs. clomiphene citrate (141 cases) to stimulate ovulation. These investigators noted that in the group of patients treated with clomiphene citrate, the risk of miscarriage in the first trimester was significantly higher, whereas the incidence of congenital malformations did not differ significantly between the two groups (letrozole, five cases (4.76%) vs. clomiphene citrate, three cases (2.121%); *p* > 0.05) [42].

In another study, Liberman et al. [40] conducted a retrospective analysis of the frequency of birth defects in newborns conceived using ART compared to those born from subfertile and fertile mothers. The study was conducted in 2004–2010 in Massachusetts, as was the study by Luke et al. [36]. Of 17,829 newborns born to mothers treated with ART, 355 had a congenital defect, whereas of 9431 live babies born to a subfertile mother, 162 had a congenital defect. On the other hand, in women with normal fertility, the birth defect was found in 6183 out of 445,080 newborns (ARR 1.5 (95% CI, 1.3–1.6) for ART and 1.3 (95% CI, 1.1–1.5) in subfertile compared with fertile deliveries. Tetralogy of Fallot and hypospadias were found to be significantly more frequent in ART newborns [40], whereas Mussa et al. [41] analyzed whether the use of ART increased the incidence of Beckwith–Wiedemann syndrome (BWS) in a population of 379,872 newborns, including 7884 ART newborns. In the group of naturally conceived babies, 31 had BWS (0.83%), whereas in the group conceived as a result of ART, there were 7 cases of BWS (8.88%; OR 10.7, 95% Cl 4.7–24.2) [41].

In a subsequent study conducted in the period from 2004 to 2016, Luke et al. [44] assessed the occurrence of malformations among 135,051 children conceived with ART, 23,647 naturally conceived ART siblings, 9396 children born to women treated with OI/IUI, and 1,067,922 naturally conceived children. A higher risk of serious, non-chromosomal birth defects was shown in ART neonates compared to those naturally conceived (ARR 1.18, 95% 1.05, 1.32), in addition to cardiovascular defects (ARR 1.20, 95% CI 1.03, 1.40) and any type of congenital defect (ARR 1.18, 95% CI 1.09, 1.27). On the other hand, when comparing the risk of a congenital defects between the group of neonates conceived with the use of ART and those conceived with the use of ICSI, the risk of a major non-chromosomal congenital defect was increased (AOR 1.30, 95% CI 1.16, 1.45 without diagnosis of male factor; AOR 1.42, 95% CI 1.28, 1.57 with diagnosis of male factor); defects of blastogenesis (AOR 1.49, 95% CI 1.08, 2.05 without male factor; AOR 1.56, 95% CI 1.17, 2.08 with male factor); cardiovascular defects (ARR 1.28, 95% CI 1.10, 1.48 without male factor; ARR 1.45, 95% CI 1.27, 1.66 with male factor). Additionally, there was an increased risk of musculoskeletal system defects (ARR 1.34, 95% CI 1.01, 1.78 without male factor) and of urogenital system defects in male infants (AOR 1.33, 95% CI 1, 08, 1.65 with the male factor) [44].

The last of the articles included in the systematic review reports our own observations. In a study carried out on a total of 1581 women, ART was used in 283 cases, and pregnancy was achieved without the use of ART methods in the remaining cases. Regardless of the applied ART method or ovulation stimulation, no statistically significant difference was found in the incidence of cardiac and non-cardiac malformations compared to naturally conceived fetuses. Out of 178 female patients in whom ovulation was induced, after which fertilization was achieved due to sexual intercourse, three developing fetuses had congenital heart defects, and there were congenital non-cardiac defects in the case of two developing fetuses. In turn, in a group of 1298 women who became pregnant without the use of assisted reproduction, congenital heart defects were observed in 11 fetuses, and congenital non-cardiac defects were detected in 32 cases. Of 105 women in whom the pregnancy was a result of extracorporeal in vitro fertilization, a congenital heart defect was diagnosed in two cases, one representing a pregnancy resulting from each of the fertilization methods (ICSI, IMSI). Furthermore, a congenital non-cardiac defect was diagnosed in two cases when fertilization was achieved through the use of sperm that had not undergone prior morphological assessment [45].

## 4. Discussion

The problem of infertility should be considered in the context of both the couple struggling with it and public health, as the inability to conceive a child causes unfavorable implications not only in terms of internal experiences but also in the partnership and in social relationships. For many couples, the decision to start infertility diagnostics and treatment is a difficult and delayed moment. Treatment of infertility is associated with fears related to the loss of intimacy, lack of control over one’s own body, complications, and therapy failure [46,47,48]. Fortunately, modern medicine offers numerous possibilities for effective diagnosis and therapy in most cases of infertility. One such possibility involves the use of ART in a situation where the current conventional and/or surgical treatment has not achieved the expected results [3,6]. Nevertheless, ART continues to be an ethical conundrum, especially in conservative societies, in addition to the fear of ART-induced birth defects [49,50,51]. In Poland, the first IVF treatment took place in 1987 at the Institute of Obstetrics and Women’s Diseases at the Medical Academy in Białystok, headed by Prof. Marian Szamatowicz [52].

This study provides a comprehensive overview of whether and to what extent the use of ART for fertilization is associated with the incidence of birth defects in children conceived through the use of ART. Ultimately, 11 original papers published in the last 10 years qualified for inclusion in the study, although the studies described therein took place during the period from 2010 to 2016. Two studies did not include information about the follow-up period [35,38]. The studies described herein concern populations in the USA [36,42,44], Canada [43], Australia [35], Japan [38,39], Iran [42], Italy [41], and Poland [45].

Only in two of the studies included in the analysis, i.e., those carried out by Tatsumi et al. [39] and Sene et al. [42], was there a control group not comprising women of normal fertility who became pregnant without the need to use ART. Their aim was to determine the safety of letrozole to stimulate ovulation either in comparison to a group of women in which ovulation was not stimulated [39] or to women whose ovulation was induced with clomiphene citrate [42]. In both studies, the induction of ovulation with letrozole or clomiphene citrate was not found to be associated with the induction of fetal malformations [39,42]. In five of studies included in our analysis, it was shown that ART significantly increases the risk of congenital malformations in this group of newborns. Nevertheless, the specifics of a given study, as well as potential confounding risk factors, should not be ignored [35,37,40,41,44]. Davies et al. [35] and Luke et al. [36] indicated that the factor determining a higher risk of developing congenital malformations among women treated IVF is the cause of infertility and its severity, although it may also depend on other factors not included in the studies. In addition, these authors noted that for ICSI but not IVF, the increased risk of congenital malformations was maintained after adjusting for maternal age and several other risk factors [35,36]. On the other hand, Tatsumi et al. [39] did not report an increased risk of congenital malformations in patients undergoing in vitro fertilization or ICSI [39]. The observations of Boulet et al. [37] also indicate a higher risk of congenital malformations after ART, and maternal ovulation disorder was a factor that significantly increased the risk of congenital malformations in ART neonates. Moreover, they indicated that the risk increases in the case of multiple pregnancies [37]. Mussa et al. [41] also showed an approximately 10-fold higher risk of BWS after ART. It should be noted, however, that during the study period, 38 newborns out of 379,872 births were diagnosed with BWS, of which 7 were conceived with the use of ART and 31 were conceived naturally [41]. Nevertheless, considering the inclusion and exclusion criteria for this review, we did not find a similar analysis in the last 10 years. The observations of Luke et al. from 2021 [44] also indicate that the use of ART increases the risk of serious non-germline birth defects, cardiovascular defects, and other birth defects in singleton pregnancies, as well as chromosomal defects in twins. In addition, it has been shown that the risk increases with the use of ICSI, especially in the case of male infertility. For this reason, the authors concluded that ICSI should only be proposed when medically indicated [44].

Nevertheless, our review included studies showing little or no association between ART and the occurrence of congenital abnormalities [38,40,43,45]. In this context, one interesting study is the analysis by Liberman et al. [40], who stated that the risk is low but clearly increases in the case of multiple pregnancies and in the subfertile group after the use of ART [40]. In our previous original study, we also found that ART did not increase the risk of cardiac and non-cardiac birth defects in fetuses. Moreover, only our study describes the use of prenatal diagnosis in the diagnosis of defects in naturally conceived fetuses and with the use of ART [45]. Moreover, the cited articles generally did not indicate whether in vitro fertilization was achieved by the IMSI or ICSI method [43], which would certainly indicate, with greater accuracy, the potential risk of congenital malformations in the case of using ART.

Therefore, taking into account the information contained in the articles included in this systematic review, we conclude that the risk of congenital defects is not directly related to the use of the ART technique itself but also depends on the age of partners, causes of infertility, comorbidities, and the number of fetuses during a pregnancy, as well as many other factors not covered in the literature.

## 5. Conclusions

In conclusion, the use of ART itself does not appear to be associated with a higher risk of developing birth defects in the fetus but with genetic, epigenetic, and environmental factors. When analyzing the selected literature, there is an impression that the conclusions contained in these works regarding the risk of congenital defects in children after ART application do not fully correspond with all of the obtained results but may instead result from the limitations of the studies, which is a normal phenomenon that affects all studies and may be related to study design.

Thus, it is necessary to impress upon infertile couples who wish to have offspring that the use of ART should not be considered risk-free but that the benefits outweigh the risks. Further education in this field, as well as social understanding, is also required

## Figures and Tables

**Figure 1 ijerph-19-04914-f001:**
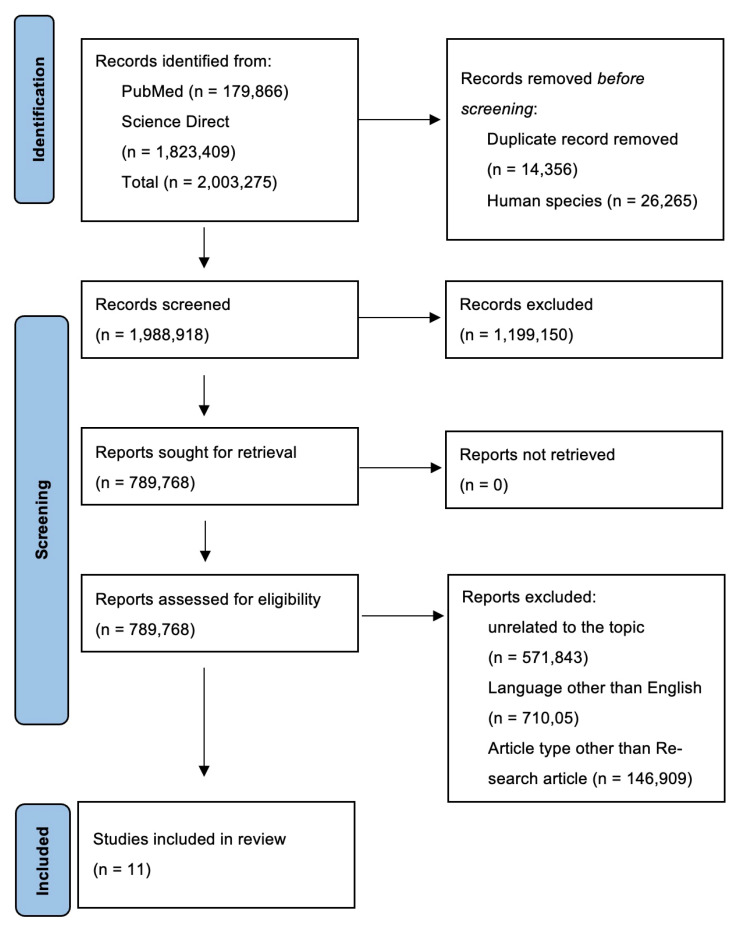
PRISMA flow diagram of systematic review. Abbreviations: PRISMA: preferred reporting items for systematic reviews and meta-analyses.

**Table 1 ijerph-19-04914-t001:** Principal recommendations for IUI [23] and in vitro fertilization [24,25,26].

Intrauterine inseminations	In vitro fertilization
Semen liquefaction dysfunctions	Irreversible fallopian tube damage
Ejaculation dysfunctions (including retrograde ejaculation)	Lack of fallopian tubes
Problems with intercourse	Endometriosis of the III or IV stages with a severity level of moderate to serious
Cervical factor	Abnormal semen in the form of severe oligoasthenoteratozoospermia or azoospermia with normal spermatogenesis
Using donor semen due to male factor infertility	Ineffective pharmacological or surgical treatment in couples with moderate male factor, idiopathic infertility, fallopian tube factor, or ovulatory dysfunctions
	Fertile couples diagnosed with recessive genetic changes for both partners, which are connected with the incidence of irreversible defects or disease in the offspring, or were diagnosed with a viral disease or encouraged to postpone fertility due to medical recommendations

**Table 2 ijerph-19-04914-t002:** Characteristics of the included studies.

Author, Year	Country, Period of Study	Type of Study	Sample Size	Main Conclusion
Davies et al., 2012 [35]	Australia (period unknown)	Original article	308,974 births (6163 after using ART; 302,811 after spontaneous conception)	High risk of birth defects after using ART
Luke et al., 2016 [36]	USA, 2004–2010	Cohort study	459,623 women (441,420 fertile, 8054 subfertile, and 10,149 IVF)	High risk of birth defects after using ART
Boulet et al., 2016 [37]	USA, 2000–2010	Original article	4,618,076 women (64,861 after using ART; 4,553,215 without the use of ART)	Increased incidence of certain birth defects in ART neonates
Iwashima et al., 2017 [38]	Japan (period unknown)	Original article	2716 pregnant women (2317 in a(SC) group and 399 (AC) group)	No link between ART and CHD
Tatsumi et al., 2017 [39]	Japan, 2011–2013	Original article	2951 women (2267 natural cycles; 684 letrozole-induced cycles resulting in pregnancy after fresh-embryo transfer)	No link between offspring after OI with letrozole and CHD
Liberman et al., 2017 [40]	USA, 2004–2010	Cohort study	472,340 live births (17,829 births after using ART; 9432 births to subfertile mothers; 445,080 births to fertile mothers)	Risk of birth defects after using ART is low
Mussa et al., 2017 [41]	Italy, 2005–2014	Original study	379,872 live births (7884 after using ATR; 371,988 without the use of ART)	ART entails a 10-fold increased risk of Beckwith–Wiedemann syndrome
Sene et al., 2018 [42]	Iran, 2007–2014	Cohort study	2009 women (1237 clomiphene citrate cycles; 772 letrozole cycles)	No link between offspring after OI and letrozole and CHD
Wen et al., 2020 [43]	Canada, 2012–2015	Original study	507,390 singleton or twin pregnancies (10,149 pregnancies assisted by ISI or IVF and 497,241 unassisted pregnancies)	No link between ART and CHD
Luke et al., 2021 [44]	USA, 2004–2013	Original study	1,236,016 children (135,051 after the use ART, 23,647 ART siblings, 9396 OI/IUI-conceived, and 1,067,922 naturally conceived)	ART is associated with increased risks of major non-chromosomal birth defects, cardiovascular defects, and any defect in singleton children and chromosomal defects in twins
Serafin et al., 2021 [45]	Poland, 2011–2016	Original study	1581 women (1298 pregnancies without the use of ART; 178 patients induced ovulation with clomiphene citrate; 137 women had intercourse naturally, 41 women AIH, 13 AID)	No link between ART and CHD and non-CHD

ART, assisted reproductive techniques; IBF, in vitro fertilization; SC, spontaneous conception group; AC, assisted conception group; ISI, intracytoplasmic sperm injection; IVF, in vitro fertilization; IUI, intrauterine insemination; AIH, artificial insemination by husband; AID, artificial insemination by donor; CHD, congenital heart defect; non-CHD, non-congenital heart defect.

## Data Availability

The data used to support the findings of this study are included in the article.

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
