# Peer review of "Evaluation of the Risk of Birth Defects Related to the Use of Assisted Reproductive Technology: An Updated Systematic Review"

_ijerph, 2022, doi:10.3390/ijerph19084914_

Round 1

Reviewer 1 Report

Comments to the manuscript: Fertility and Assisted Reproduction Techniques

Major comments.

  • It is necessary to considerate if IJERPH is the right journal for this paper, since the aim of IJERPH is “to address critical issues related to environmental quality and public health”.
  • A major modification of the manuscript is needed in order to be published in IJERPH, since the topics that are reviewed namely 1) The problem of Infertility, 2) Causes, Diagnosis, and treatment of infertility, 3) Assisted Reproduction Technics, and 4) Prenatal Diagnosis, have been recently reviewed in the articles below and others. It is necessary to highlight the originality of the paper. What makes it different from the previously reported reviews?

Vander Borght M, Wyns C. Fertility and infertility: Definition and epidemiology. Clin Biochem. 2018 Dec;62:2-10

Carson SA, Kallen AN. Diagnosis and Management of Infertility: A Review. JAMA. 2021 Jul 6;326(1):65-76.

Szamatowicz M, Szamatowicz J. Proven and unproven methods for diagnosis and treatment of infertility. Adv Med Sci. 2020 Mar;65(1):93-96.

Jack Yu Jen Huang , Zev Rosenwaks. Assisted reproductive techniques. Methods Mol Biol 2014;1154:171-231.

Carlson LM, Vora NL. Prenatal Diagnosis: Screening and Diagnostic Tools. Obstet Gynecol Clin North Am. 2017 Jun;44(2):245-256

  • The review lacks of figures/tables to make the text easier to read.

Minor comments.

-The title needs to be modified, since prenatal diagnosis is reviewed in addition to fertility and assisted reproduction techniques.

-Lines 75-77: Please note that the prenatal diagnosis has not been included in the aim of the manuscript.

-Line 16: Please rephrase “Although the last thirty years of progress as far as treatment of infertility with the use of medical methods of assisted reproduction caused this method to become common practice”.

-Line 25: Please replace “bases” for “basis”

-Line 30: It is not clear in the text in which situation there should be a consideration whether to supplement the diagnosis based on invasive methods.

-Line 33: Please list the keywords

Reviewer 2 Report

This is an academic review on the subject of fertility and assisted reproductive techniques.
It is a structured summary of the techniques used, but it does not provide a systematic review or any study of any kind, and therefore does not provide evaluable results.

As an academic review, it does not provide relevant information for clinical use.

Author Response

“This is an academic review on the subject of fertility and assisted reproductive techniques.
It is a structured summary of the techniques used, but it does not provide a systematic review or any
study of any kind, and therefore does not provide evaluable results. As an academic review, it does not
provide relevant information for clinical use.”
We would like to thank the Reviewer’s for this comment. According to the recommendation, we have
changed our paper from an academic review to a systematic review. We hope that we included new data
and valuable results from studies.
We thank the Editor and the Reviewers for helpful comments, and for recognizing the value of our study.
We would like to assure that the manuscript was read and carefully corrected by English native speaker
once again.
We hope that the improved work will meet your expectations.
Thank you for your consideration of this manuscript.

Round 2

Reviewer 1 Report

The reviewers comments have been adressed, and the manuscript has been significantly improved..